# Perceptions of persons deprived of liberty regarding tuberculosis vaccine research

Mariana Cristina Campos Falleiros Pires[1], Yiran E. Liu[2], Everton Ferreira Lemos[3], Liliane Ferreira da Silva[1], Mariana Garcia Croda[1], Monica Magalhães[4], Dhélio Batista Pereira[5], Mariana Pinheiro Alves Vasconcelos[5], Rosilene Ruffato[5], Solana Monteiro Batista[5], Giselle Lima de Freitas[6], Marcelo Cordeiro-Santos[7], Giane Zupellari dos Santos-Melo[8], Jair dos Santos Pinheiro[7,8,9], Lara B. O. Assis[9], Lia Gonçalves Possuelo[10], Tiago Antonio Heringer[10], Daiane Kist Back[10], Pauline Schwarzbold[11], Jason R. Andrews[2], Crhistinne Cavalheiro Maymone Gonçalves[1], Julio Croda[12]*

1 School of Medicine, Federal University of Mato Grosso do Sul, Campo Grande, Brazil, 2 Division of Infectious Diseases and Geographic Medicine, Department of Medicine, Stanford University, Stanford, California, United States of America, 3 State University of Mato Grosso do Sul, Campo Grande, Brazil, 4 Pensi Institute, São Paulo, Brazil, 5 Tropical Medicine Research Center, Porto Velho, Rondônia, Brazil, 6 School of Nursing, Federal University of Minas Gerais, Belo Horizonte, Brazil, 7 Tropical Medicine Foundation Doctor Heitor Vieira Dourado, Manaus, Brazil, 8 University of the State of Amazonas, Manaus, Brazil, 9 Amazonas Health Surveillance Foundation, Manaus, Brazil, 10 University of Santa Cruz do Sul, Santa Cruz do Sul, Brazil, 11 Criminal Police of Rio Grande do Sul, Santa Cruz do Sul, Brazil, 12 Oswaldo Cruz Foundation, Campo Grande, Mato Grosso do Sul, Brazil

* juliocroda@gmail.com

## Abstract

Several tuberculosis (TB) vaccine candidates are currently advancing to late-stage clinical trials. Prisons in low-and middle-income countries harbor some of the highest rates of TB in the world, making persons deprived of liberty (PDL) an important population to prioritize for the introduction of effective vaccines. However, the inclusion in clinical trials raises significant ethical concerns due to a history of exploitation and mistreatment within medical research. To date, PDL own perspectives on participating in vaccine research have been largely overlooked. This multicenter qualitative study employed focus group (FG) discussions in seven state prisons across four of the five regions of Brazil, involving 91 incarcerated individuals (64 men and 27 female) between February and August 2024. The discussions explored participants' perceptions regarding health in prisons, TB, vaccines in general, new TB vaccines, and their potential participation in research. Participants reported encountering difficulties in accessing healthcare services within the prison system and shared personal or indirect experiences with TB, as well as concerns about their family members being at risk for TB exposure. While they generally held positive perceptions about vaccines and trials, they emphasized the need for clear and transparent information, respect for individual autonomy, and assurances of accountability from researchers as conditions of their willingness to participate in future trials. PDL perceptions regarding

**Data availability statement:** The data that support the findings of this study are publicly available at https://github.com/juliocroda/Perceptions-of-persons-deprived-of-liberty-regarding-tuberculosis-vaccine-research . This dataset consists of verbatim statements from participants, which have been subjected to a thematic coding process. It is important to note that the same statement may be relevant and, therefore, simultaneously classified in more than one coding category. This approach aims to capture the complexity and interconnected-ness of the themes addressed by incarcerated individuals, ensuring that the richness of their perceptions about health, autonomy, risk, and trust is fully preserved in the sub-codings.

**Funding:** This work was funded by the Open Philanthropy Project Fund, a donor-advised fund of the Silicon Valley Community Foundation (Grant Number: 2023-333629 (5384) GB-1602079), through a research grant administered by the Fundação para o Desenvolvimento Científico e Tecnológico em Saúde (Fiotec), for authors MCCFP, EFL, LFS, RR, GLF, LBOA, and LGP. The funders had no role in study design, data collection and analysis, decision to publish, or preparation of the manuscript. No author received a salary from any of the funding entities (Open Philanthropy Project Fund/Silicon Valley Community Foundation and Fiotec) or from commercial companies.

**Competing interests:** The authors have declared that no competing interests exist.

participation in clinical trials for new TB vaccines are significantly influenced by their prior experiences with the prison health system and their level of trust in research institutions. To ethically and effectively include PDL in future research, it is crucial to prioritize respect for autonomy and transparent communication about the risks and potential benefits involved.

## Author summary

This study explores the perspectives of incarcerated individuals in Brazil about TB, vaccines, and participation in clinical trials. Despite barriers to health access and trust, many showed openness to research, highlighting the need for transparent communication, respect for autonomy, and ethical safeguards. These insights offer valuable guidance for designing inclusive and ethically sound vaccine trials involving PDL.

## Introduction

Tuberculosis (TB) remains a significant public health concern worldwide, causing an estimated 1.25 million deaths and 10.8 million new cases in 2023 [1]. It disproportionately impacts persons deprived of liberty (PDL), particularly in Latin America, where prison populations have grown by nearly 300% in the past three decades [1,2]. The conditions of imprisonment, such as overcrowding and malnutrition, increase the risk of PDL acquiring TB infection and disease. In South America, the risk of TB is 26 times higher in prisons compared to the general population [3]. Furthermore, TB in prisons can spill over into communities through staff, visitors, and individuals released from incarceration, who have an elevated risk of TB for up to seven years post-release [4,5]. A recent mathematical modeling study found that incarceration is a significant driver of population-level TB in Latin America, accounting for an estimated 27% of new cases, more than any other risk factor [6]. In addition, a genomic epidemiologic study conducted in Brazil found that up to 50% of community-based TB cases may originate from transmission within prison settings [7]. Ending the TB epidemic will require comprehensive efforts to address TB in prisons.

To date, interventions to reduce TB in high-burden prisons have been lacking. Structural reforms such as reducing incarceration rates and overcrowding are critical for mitigating transmission [6] but present political challenges often beyond the purview of national TB programs. Systematic screening for TB in prisons, now strongly recommended by the World Health Organization (WHO), has demonstrated high yield in empirical studies in Latin American prisons [8]. However, serial rounds of annual mass screening in three Brazilian prisons did not significantly reduce TB prevalence, suggesting the need for more frequent screening and complementary measures [9]. Tuberculosis preventive therapy (TPT) presents as a promising strategy [10], but evidence of its effectiveness in high-transmission prisons remains limited [11].

To meaningfully reduce TB in prisons, new tools, such as effective vaccines, may be needed to complement existing measures. The only licensed TB vaccine, BCG, was discovered a century ago and is inadequate in preventing TB in adolescents and adults [12]. The current TB vaccine development pipeline has several promising candidates, including M72/AS01, MTBVAC, and VPM1002, which have advanced to phase 3 trials after demonstrating favorable safety profiles and preliminary evidence of efficacy [13–15]. However, TB vaccine trials face significant challenges. The relatively low incidence of TB, even in high-burden community settings, necessitates the recruitment of thousands of participants and prolonged follow-up periods to achieve adequate statistical power. This results in slow and costly trial processes that hinder rapid advancements in vaccine development.

PDL may significantly benefit from effective TB vaccines and are recognized as one of many high-risk, target populations to be prioritized for rapid access to new TB vaccines once licensed [16]. However, in contrast to other priority populations—like persons living with HIV, persons with diabetes, and persons experiencing malnutrition—PDL have been systematically excluded from participation in TB vaccine trials. Including PDL in vaccine trials presents serious ethical challenges. Historically, PDL have been subjected to exploitative, non-consensual experimentation [17–19]. Barriers to privacy, autonomy, and access to healthcare in prison further complicate ethical trial implementation. During the COVID-19 pandemic, as PDL experienced elevated rates of COVID-19 infection and mortality, debates emerged around the risks and benefits of including PDL in COVID-19 vaccine trials, and the necessary conditions to make ethical participation possible [20–23]. Despite agreement on the critical importance of engaging PDL in these debates, the perspectives of PDL toward participation in vaccine trials, or scientific research more broadly, remain poorly understood, especially outside of high-income countries. This omission is particularly striking given that tuberculosis remains the leading cause of death from an infectious disease among adults worldwide [24,25], and that recent perspectives have underscored both the injustice of global TB disparities [25] and the ethical imperative to evaluate ultra-short TB prevention regimens with careful scientific and ethical scrutiny [26].

In Brazil, where the worsening TB epidemic is increasingly driven by incarceration, understanding the experiences and perspectives of PDL will be crucial to inform effective, ethical, and patient-centered strategies to accelerate TB elimination. In this study, we conducted qualitative focus group (FG) discussions in seven male and female prisons in Brazil to understand perceptions toward TB, TB research, and participation in TB vaccine trials.

## Materials and methods

### Ethics statement

The study was approved by the Ethics Committee for Research (CEP) of the Federal University of Mato Grosso do Sul – UFMS (Approval No. 6.612.127, CAAE: 75865823.7.1001.0021, January 12, 2024), the Center for Research in Tropical Medicine of Rondônia – CEPEM (Approval No. 6.633.785, CAAE: 75865823.7.2001.0011, February 3, 2024), the School of Nursing at the Federal University of Minas Gerais – UFMG (Approval No. 6.708.897, CAAE: 75865823.7.2002.5149, March 18, 2024), the University of Santa Cruz do Sul – UNISC (Approval No. 6.711.854, CAAE: 75865823.7.2005.5343, March 19, 2024), the Tropical Medicine Foundation Doctor Heitor Vieira Dourado – FMT-HVD (Approval No. 6.745.992, CAAE: 75865823.7.2004.0005, April 5, 2024), and Stanford University (IRB #76262) as well as by the respective Brazilian State Departments of Penitentiary Administration.

All procedures complied with the ethical principles outlined in the Declaration of Helsinki and the Brazilian National Health Council Resolution No. 466/2012.

### Study setting

In this multicenter qualitative study, we conducted FG discussions in five male prisons and two female prisons in Brazil between February and August 2024. The participating prisons were located in Campo Grande (state of Mato Grosso do Sul), Manaus (Amazonas), Montenegro (Rio Grande do Sul), Porto Velho (Rondônia), and Ribeirão das Neves (Minas Gerais) (Fig 1).

## Map of Brazil with Participating Centers

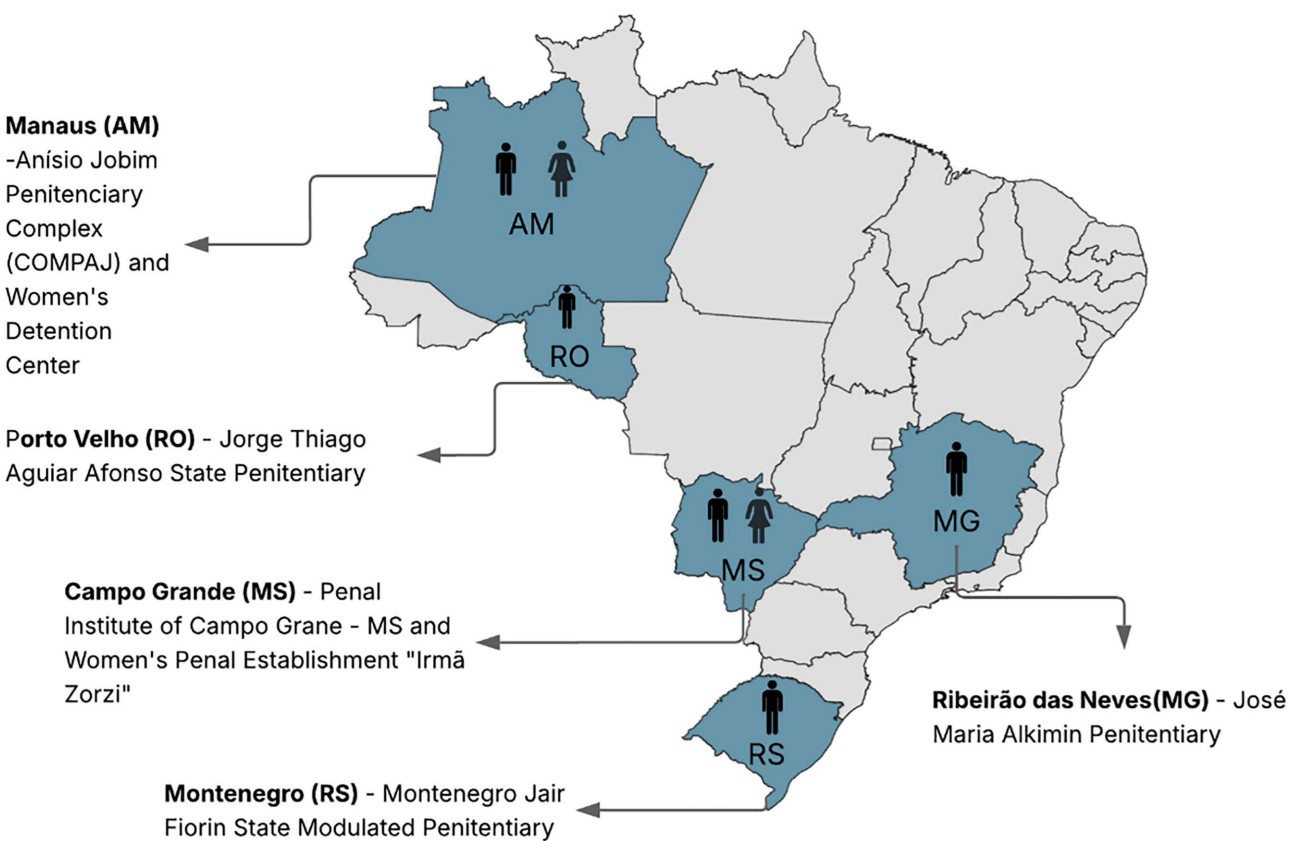

**Manaus (AM)**
-Anísio Jobim Penitenciary Complex (COMPAJ) and Women's Detention Center

**Porto Velho (RO)** - Jorge Thiago Aguiar Afonso State Penitentiary

**Campo Grande (MS)** - Penal Institute of Campo Grane - MS and Women's Penal Establishment "Irmã Zorzi"

**Montenegro (RS)** - Montenegro Jair Fiorin State Modulated Penitentiary

**Ribeirão das Neves(MG)** - José Maria Alkimin Penitentiary

**Fig 1. Map of Brazil with study centers. Legend:** Geographic distribution of study recruitment sites across Brazil. Recruitment for the study took place from February 15 to August 6, 2024, non-simultaneously at all centers: Federal University of Mato Grosso do Sul – UFMS (February 15 to March 23, 2024), Center for Research in Tropical Medicine of Rondônia – CEPEM (April 15, 2024), School of Nursing at the Federal University of Minas Gerais – UFMG (May 17, 2024), Medicine Foundation Doctor Heitor Vieira Dourado – FMT-HVD (July 25–26, 2024), and University of Santa Cruz do Sul – UNISC (August 6, 2024). The base layer shapefile for Brazil's administrative boundaries was obtained from the Brazilian Institute of Geography and Statistics (IBGE) and is publicly available at https://www.ibge.gov.br/geociencias/organizacao-do-territorio/malhas-territoriais/15774-malhas.html. Use of the IBGE cartographic base complies with the CC-BY 4.0 license and the institution's open data terms of use.

### Participant recruitment and eligibility criteria

Participants (N = 91) were recruited through a multi-stage convenience sampling strategy, designed to ensure inclusion of both male and female incarcerated individuals from diverse cells and pavilions across participating prisons. Prior to recruitment, research teams from the participating center conducted visits to each participating prison to (i) present the study to prison administrators and health/social service staff, (ii) identify appropriate spaces for focus group discussions, and (iii) discuss eligibility criteria and logistical considerations with prison social service teams.

Given the security restrictions within prison settings, potential participants were initially identified by the prison social service sector in each unit. Identification followed the predefined eligibility criteria, with a special emphasis on ensuring representativeness, but was also subject to operational considerations (e.g., movement schedules, security clearance). Eligibility criteria included: (1) age 18 years or older; (2) literacy; (3) a cumulative incarceration period of at least 12 months (continuous or non-continuous); and (4) provision of voluntary informed consent.

A total of 93 individuals were initially approached across all sites. Upon arrival at the focus group setting, the research team explained the study objectives, procedures, potential risks and benefits, and emphasized voluntariness. The Free and Informed Consent Form (Termo de Consentimento Livre e Esclarecido – TCLE) was read aloud and discussed with the PDL. Individuals were explicitly asked whether they wished to stay or return to their cells without any consequences. Two individuals declined participation at this stage, choosing to return to their cells. No reasons for refusal were provided. No participants withdrew after signing informed consent. Those who agreed to participate signed two copies of the form—one was kept by the participant or filed in the medical record.

## Study procedures

To develop the FG question script (S1 Table), we adapted the Vaccine Hesitation Matrix (VHM), developed by the WHO Strategic Advisory Group of Experts on Immunization [27]. The script was divided into three main dimensions, which were further subdivided into two categories each:

1. perceptions of health in the prison (prison health services, experiences with TB)

2. perceptions of vaccines (vaccines in general, new TB vaccines)

3. perceptions of participation in research (clinical trials for a new TB vaccine, PDL autonomy).

The full list of FG questions is provided in the Appendix.

Each focus group was conducted by two facilitators and two observers, in private settings within the prisons. The facilitators were experienced in working in prison environments, qualitative health research, including focus group discussions. Observers took field notes during each focus group.

All focus groups followed a standardized structure. Participants were first informed of the study's objectives and requirements. The facilitators read the Informed consent form aloud, addressing questions from potential participants. Those who decided to participate signed the consent form.

Next, study participants were interviewed individually in private areas using a structured questionnaire designed to collect clinical and epidemiological data. Variables investigated included: sex, age, race/ethnicity, marital status, education level, household income, city/state of residence before incarceration, comorbidities, risk factors, duration of incarceration, transfers between prisons, visitation status, history of contact with a TB patient in the same cell, current TB treatment status, past TB treatment, and use of medications.

The focus group discussion then began, with audio recording. Each group consisted of 12–15 individuals and lasted approximately 90 minutes on average. To protect confidentiality, participants were assigned random numbers and instructed to identify themselves by their assigned number before speaking. Any names mentioned during the sessions were removed from the transcripts.

## Data processing and analysis

Questionnaire data were stored in a secure REDCap server with controlled access. Audio recordings of FG discussions were stored in a secure password-protected folder accessible only to the research team. Transcriptions of FG discussions were performed verbatim by two researchers, maintaining authenticity and literal accuracy. The data were analyzed using the thematic content analysis method proposed by Minayo (2005) [28], following three steps:

1. Pre-analysis: Thorough and detailed reading of the transcripts.

2. Material exploration: Extracting and coding key segments of speech, and grouping them into thematic categories based on recurrence, relevance, and uniqueness.

3. Result processing: Synthesizing and interpreting themes using updated theories and scientific literature to draw robust and relevant insights.

The transcriptions were reviewed and coded by two coders and grouped by thematic areas throughout the analysis, resulting in the emergence of specific dimensions, categories, and subcategories. Triangulation was achieved by holding interactive discussions with facilitators and observers and sharing preliminary results in research team meetings to validate the coding. MaxQDA24 [29] software was used to organize and highlight the coded texts.

## Results

We conducted a total of seven focus groups with 91 PDL, of which 64 (70%) were men and 27 (30%) women. Participants had an average age of 35 years and many had low education levels (41% with no high school) (Table 1). Half of participants identified as mixed race (50%), followed by white (25%) and black (20%). Regarding duration of incarceration, 41% had been imprisoned for up to 2 years, 18% for 3–4 years, 24% for 5–9 years and 18% for 10 years or more. The majority of participants (63%) had a history of prior incarceration, 66% had been transferred between prisons, and 67% reported receiving visitors. Among participants, 20% had a prior TB diagnosis, 3% were currently undergoing TB treatment, and 60% reported having had contact with cellmates who had TB. In the prison units where focus groups were conducted, the reported incidence of TB varied greatly, ranging from 685 to 4,533 per 100,000 person-years.

Participating PDL described their perceptions regarding the health system in prisons, vaccines in general, a potential new TB vaccine and participation in clinical trials. Representative quotes are presented in Table 2, organized by dimensions, categories, and subcategories that emerged in the analysis. The detailed distribution of these narratives by state and prison type (male/female) can be consulted in (S2 Table). Although most perceptions were consistent across the groups, some regional or gender variations were observed. The matrix of narrative summary, detailing the convergence and divergence by core theme, is presented in (S3 Table). Both male and female participants converged on systemic barriers to health access, citing staff negligence and the difficulty of physical access to the health unit as primary issues.

Participants expressed a spectrum of attitudes toward participating in a potential tuberculosis vaccine trial. While several viewed research as an opportunity to improve health care, contribute to science, and protect their peers and families, others expressed hesitation or outright refusal. Willingness was largely conditional on prior testing, transparent communication, and assurance of safety. Some participants emphasized trust in external research teams rather than prison staff, perceiving the former as more respectful and reliable. However, others voiced fears of being treated as "guinea pigs," recalled adverse experiences with previous vaccines, and resisted the idea of studies conducted exclusively among incarcerated populations, which they associated with coercion and inequality. Thus, participation was framed both as an act of solidarity and as a risk of exploitation.

### Perceptions about health in prison

**Access to health care in prison.** Participants described many challenges in accessing health care in prison (Table 2). Access to essential medicines was limited, as one participant described: "Sometimes you need insulin and there is no insulin. I ended up staying in bed for seven days because there was no insulin. My wife had to buy it. And it takes a week for us to make a call to our family so they can send the medication."

Participants also reported significant difficulties in accessing health services, in part due to factors external to the health team. In particular, participants cited logistical and structural barriers such as clinic hours of operation and the lack of guard escorts to the health unit. As one participant explained, "The health service is very good…but the difficult part is getting there…leaving the cell, that's the hardest part for the prisoner." Access to health services could be further hindered by the attitudes and attentiveness (or lack thereof) of prison staff. As one participant put it, "Honestly, there is a lot of negligence on the part of some of the employees, some try to help us, but others, for them, it doesn't matter, they just want to come in, finish their day, and leave."

**Perceptions and experiences of TB.** Participants described TB as a serious, potentially fatal disease causing various physical symptoms and requiring rigorous treatment. Accordingly, participants frequently cited fears of developing TB and

**Table 1. Clinical and epidemiological characteristics of focus group participants (N=91).**

| Characteristic | n (%) |
|---|---|
| **Sex** | |
| Female | 27 (30) |
| Male | 64 (70) |
| **Age group** | |
| 20-39 years | 62 (68) |
| 40-59 years | 27 (30) |
| 60+years | 2 (2) |
| **Race** | |
| Asian | 4 (4) |
| White | 23 (25) |
| Indigenous | 1 (1) |
| Mixed | 45 (50) |
| Black | 18 (20) |
| **Education** | |
| Incomplete elementary education | 24 (26) |
| Complete elementary education | 13 (14) |
| Incomplete high school | 22 (24) |
| Complete high school | 16 (18) |
| Incomplete higher education | 10 (11) |
| Complete higher education | 2 (2) |
| Postgraduate | 1 (1) |
| Did not respond | 3 (3) |
| **Length of incarceration** | |
| Up to 2 years | 37 (41) |
| 3-4 years | 16 (18) |
| 5-9 years | 22 (24) |
| 10 years or more | 16 (18) |
| **Transfers between prisons** | |
| Yes | 60 (66) |
| **Previous incarceration** | |
| Yes | 57 (63) |
| **Receive visits** | |
| Yes | 60 (67) |
| **TB history** | |
| Yes | 18 (20) |
| **Contact with a cellmate with TB** | |
| Yes | 55 (60) |
| **Undergoing TB treatment** | |
| Yes | 3 (3) |

the pervasiveness of the disease in prison. One participant described the perceived inevitability of exposure as follows: "Here in prison, there is no way for us to get rid of the problem, right? Whether you like it or not, you will have direct contact with [a person with TB], and we can end up getting infected."

Participants who had personal or secondhand experience of TB reported feelings of dismay upon diagnosis and distress from witnessing the suffering of others with TB. One participant reflected on the moment she was told she had TB: "I

**Table 2. Perceptions of participants regarding health care in prisons, TB, vaccines, and participation in research.**

| |
| --- |
| **Perceptions about TB and health care in prison** |

*Limited access to health needs"*
"Sometimes you need insulin and there is no insulin. I ended up staying in bed for seven days because there was no insulin. My wife had to buy it. And it takes a week for us to make a call to our family so they can send the medication. So it's terrible". **(Female)**

*Negligence of prison staff*
"Honestly, there is a lot of negligence on the part of some of the employees, some try to help us, but others, for them, it doesn't matter, they just want to come in, finish their day, and leave". **(Male)**

*Difficulty in accessing the health services*
"'And depending on the time, there is no escort. [...]. They don't even open the little window [at the door of the cell], they tell everyone, leave it alone, don't mess with it, tomorrow we'll see what we'll do''. **(Female)**
"The health service is very good and they provide very good service, but the difficult part is getting there, to the health service, leaving the cell, that's the hardest part for the prisoner". **(Male)**

*Personal or second hand experience of TB; seriousness of disease*
"So, it was very sad to see her suffering, you know? The bloody sputum, the thinness, the lungs, the daily shortness of breath, the entire situation, of deploring the body [...]". **(Female)**
"No, I got there and went in, the nurse said, I'm going to tell you something, you have tuberculosis. And then, like, I started crying, it was like I wasn't walking on the ground [...]. Who expects illness in their life?". **(Female)**
''I think that people die even with treatment, because the disease requires attention, because it is a serious disease." **(Male)**

*Fear of getting TB and spreading to others*
"I was afraid of getting involved with my wife because of tuberculosis. Trying to deprive her, let this period of fifteen days pass [...]". **(Male)**

*Lack of information*
"[...], I myself am afraid of catching tuberculosis. Because, especially here in prison, there is no way for us to get away from the problem, right? Because sometimes, [it's?] the cellmate who is living in the same cell or pavilion, on the same side of the bathroom, on the same side when sunbathing. Whether you like it or not, you will have direct contact with him, and we can end up getting infected". **(Male)**
"I never had any information about this disease. I don't know, I don't even know if it's a bacteria or a virus. Normally I don't have any information". **(Female)**

| |
| --- |
| **Perceptions about vaccines** |

*Positive perceptions toward vaccines*
"I think it's really prevention. Because I have a son, right. Since my son was born, I have been giving him medication [vaccines]. It's important that we take this care in the past, because the future is coming. It's also important for people to study for this. And specialize in this area. For the prevention of humanity, right. Our health is important. It's very, very important''. **(Female)**
"So, I think there should be more vaccines for people with intimate diseases, right, that... gynecological. So, because they often go through a lot... It's just a pill. I think the vaccine... It gives a faster result. Do you understand?". **(Female)**
"If there is a vaccine, with explanations, with everything in order, everyone will embrace the cause. Everyone wants to go home. No one wants to go and infect anyone. We don't want to be a disappointment". **(Male)**
"We have to thank that woman who discovered the [COVID-19] vaccine. She helped millions of people not die." **(Male)**

*Lack of information/explanation about vaccination*
"At a certain point, they arrive at the prison, call the whole pavilion to go to get vaccinated, and they don't explain what the situation is with the vaccine''. **(Male)**

*Side effects from vaccination*
"There are people who get the flu, fever and body aches when they get vaccinated." **(Male)**

*Positive perceptions about new TB vaccines and vaccine research*
"I would be proud to know that the government is going to do this [test new vaccines], that it is betting on this, that doctors and researchers have invested in this". **(Female)**
"The creators [of the vaccine], the people who spent time studying this should explain this to people nationally." **(Female)**
"You have to take it, [...] the researcher is not going to keep making vaccines, making vaccines, making vaccines. When you make the vaccine, you're not going to do it to harm the person, you're not, you're doing it for the good of others, you understand? For me, you have to take the vaccine". **(Male)**

*Understanding of potential limitations and uncertain efficacy"*
"But the vaccine doesn't mean you won't get it, you will get it but with less symptoms". **(Female)**

| |
| --- |
| **Perceptions about participation in research** |

Belief in significance and desire to contribute
"I think that I will participate in this study, which is important for humanity, because the vaccine will last forever, it will be passed down from generation to generation, and it will serve your grandchildren and great-great-grandchildren." **(Female)**

*Conscious and informed participation*
"I would accept being part of this research, especially because the vaccine is something very important for our human society, as we know that since the last century there have been many diseases whose mortality has been avoided precisely because of the vaccine [...]." **(Male)**

*(Continued)*

**Table 2.** (Continued)

| Perceptions about participation in research |
| --- |
| *Conscious and informed participation* |
| '[...] we understand that we will participate in the research and we may be guinea pigs. But we will have to be well aware of what we are doing". (**Female**) |
| "I think the vaccine could start inside the prisons, (...) if they came here, brought everyone together and reached a consensus [...] It's not just about coming here without giving any explanation and simply saying, let's vaccinate everyone for tuberculosis". (**Female**) |
| **Negative perceptions and barriers to trust in health research** |
| I wouldn't participate, I don't want to be anyone's guinea pig. (**Male**) |
| "For me, if it's a test, it's risky. I'm scared. I've seen people die from medicine that was supposed to cure them."(**Female**) |
| "If it's only to be tested on prisoners, then yes, [I would be] a guinea pig. I would only accept it if it were for everyone.**" (Male)** |
| "I wouldn't take it, because I don't know what could happen, and nobody here takes responsibility. (**Female**) |
| *Right to access and make one's own decision about participation* |
| "And then, we also have the same rights as any citizen, other than being able to return to our home, right?". (**Female**) |
| ''Yes, because for example, whether I participate in this study is my decision, whether I go or not, even though I'm incarcerated. Because it's my health, right? So, it's the person's own decision. I may participate and she may not, I'll go because I feel safe." (**PU Female**) |
| "Just as it's doing us good, it will also be doing our family good and we're having contact through the visit, so it would be good! [...] I take it [the vaccine] here, she takes it [the vaccine] there". (**Male**) |
| *Transparency and accountability for risks* |
| "But, if you gave us the explanation, the paper here [informed consent form], which we will be signing and reading, and it said there that you would be responsible for anything serious or any mistake that happened to us, then yes, I would participate". (**Male**) |
| *Respect for autonomy* |
| "I think everyone felt comfortable here, because from the beginning of the conversation it started the right way, they [the focus group facilitators] started by asking who wanted and didn't want to stay. And they gave us the chance, do we want to stay or not?". (**Male**) |
| "It's mandatory to take it [Must get vaccinated against COVID-19]. Even if someone refuses, if they insist on not taking it, they might lose, lose their visitation rights — as we've already been threatened with losing visits. [...] That's why it becomes an obligation". (**Male**) |

started crying…it was like I wasn't walking on the ground." Another participant with previous TB expressed concern about transmitting TB to a family member during visits.

**Lack of information.** While participants had a basic understanding of TB as an infectious disease, many reported the difficulty of obtaining accurate information about TB. They highlighted the lack of access to reliable information sources and the absence of educational activities in prison. Some expressed uncertainty about the causative agent of TB, routes of transmission, and effective preventive measures.

## Perceptions of vaccines

**Positive sentiments toward vaccines.** Vaccines were generally viewed favorably by participants, with wide recognition of the importance of vaccination for protecting the health of oneself and one's family. Scientific advances in immunization, especially for vaccines against COVID-19, were lauded by participants as historic, lifesaving milestones. Some participants also called for the development of vaccines against other diseases, expressing a preference for prevention over treatment.

Regarding the development of new vaccines against TB, participants applauded the efforts of researchers and governments to seek solutions to control the disease, viewing it as a social good for humanity. As one participant stated, "I would be proud to know that the government is going to do this…that doctors and researchers have invested in this." Another participant expressed confidence in the benevolent motivations of those developing the vaccine: "When you make the vaccine, you're not going to do it to harm the person…you're doing it for the good of others."

**Perceptions about side effects and efficacy.** Participants expressed concerns about possible side effects from vaccination, citing first- or second-hand experience of flu-like symptoms, fever, and body aches. They also reported unease around potential unknown adverse reactions and the lack of follow-up after vaccination. Some participants

described the potential limitations of vaccines, recognizing that vaccines may not offer full protection but could reduce the severity of disease.

**Negative experiences with vaccination in prison.** Some participants expressed dissatisfaction or discomfort with prior vaccination efforts within the prison. For instance, one participant recalled an experience of being called to get vaccinated without clear information or understanding: "They arrive at the prison, call the whole pavilion to go to get vaccinated, and they don't explain what the situation is with the vaccine."

## Perceptions about participation in research

**Perceptions toward participation in clinical trials.** Many participants expressed a desire to participate in clinical trials for a new TB vaccine, describing it as an opportunity to contribute to an effort they viewed as significant for society. One participant stated, "I think that I will participate in this study, which is important for humanity, because the vaccine will last forever, it will be passed down from generation to generation, and it will serve your grandchildren and great-great-grandchildren."

Some participants associated participation in clinical trials with the idea of being "guinea pigs", especially if the trials exclusively enrolled PDL: "If it's only to be tested on prisoners, then yes, [I would be] a guinea pig. I would only accept it if it were for everyone." Others expressed concerns about the safety risks of participating in clinical trials: "If it's a test, it's risky. I'm scared. I've seen people die from medicine that was supposed to cure them." Yet, many others expressed a willingness to participate nonetheless, as long as it was with intention and awareness of potential risks and benefits: "We understand that we will participate in the research and we may be guinea pigs. But we will have to be well aware of what we are doing." Participants gave a variety of reasons for their decision to participate, ranging from a desire for personal benefit and altruism to concerns about the risks of the study.

**Information, transparency, and accountability for risks.** Participants highlighted detailed information and transparent communication as crucial factors influencing their participation in research. In particular, they called for clear explanations of potential risks and benefits, as well as reliable mechanisms for protection and recourse in the case of unanticipated harms. One participant said, "If you gave us the explanation, the paper here [informed consent form], which we will be signing and reading, and it said there that you would be responsible for anything serious or any mistake that happened to us, then yes, I would participate." Another participant described the importance of dialogue and engagement with PDL to achieve consensus before conducting vaccine trials in prisons: "The vaccine could start inside the prisons, (...) if they came here, brought everyone together and reached a consensus. It's not just about coming here without giving any explanation, simply saying, let's vaccinate everyone for tuberculosis".

**Right to decide to participate.** A central theme in the focus groups was the declaration of one's right to make decisions about participating in research. One participant advocated for autonomy in making decisions that would affect her own health: "Whether I participate in this study is my decision…even though I'm incarcerated. Because it's my health, right?" Another participant claimed the right to participate in research as a human right to which she, "like any citizen," was entitled, despite being deprived of the right to liberty ("to return to our home"). This sentiment was corroborated by yet another participant who alluded to a scenario wherein incarcerated individuals and community members alike were eligible to participate in vaccine trials: "Just as it's doing us good, it will also be doing our family good…I take it [the vaccine] here, she takes it [the vaccine] there." Such attitudes were prevalent throughout the focus groups, revealing a strong consensus around PDL's perceived ability and right to make informed decisions around participation in research.

**Respect for autonomy and its challenges in prisons.** Participants reported positive experiences participating in the focus group, emphasizing that their freedom of choice was respected. As one participant expressed, "I think everyone felt comfortable here, because from the beginning the conversation started in the right way... they gave us the chance, do we want to stay or not?" This approach was perceived as honoring each individual's autonomy, fostering an atmosphere of trust and respect.

At the same time, other accounts revealed that autonomy may be compromised under certain conditions. While participation is often presented as voluntary, some individuals described feeling indirect pressure or fear of consequences for refusing—such as threats to visitation rights. One participant, referring specifically to the context of the COVID-19 pandemic and the vaccination campaign within the prison, stated: "It's mandatory to take it. Even if someone refuses, if they insist on not taking it, they might lose, lose their visitation rights — as we've already been threatened with losing visits. [...] That's why it becomes an obligation."

### Recommendations for ethical inclusion of PDL in clinical trials of new TB vaccines

Drawing from the perspectives gathered in the FG, we propose a set of stakeholder-informed recommendations to guide the inclusion of PDL in clinical trials for new TB vaccines. These recommendations are not intended to be an exhaustive list of necessary conditions; rather, they represent responses to key points raised by PDL, whose voices and perspectives are too often excluded from the research process. The proposed actions include:

1. Ensure Universal, Non-Coercive Access to Quality Healthcare

Ensure universal access to quality health care in prison before, during, and after the trial, for all PDL regardless of trial participation. Substandard care may introduce coercive incentives to participate in research as a means to access better care. Vaccine trials must not exploit or exacerbate existing inequalities in the prison health system.
Actionable Steps:

- Conduct an independent audit of medical care in putative trial sites. Ensure that adequate standards of care, including access to essential medication, are met before initiating trial enrollment.

- Ensure adequate staffing in putative trial sites so that limited availability of staff escorts does not impede access to medical care. Assess additional staffing needs for trial implementation and ensure that trial activities do not pose undue burden on routine activities in the prison, including routine medical care for non-trial-related needs, staff escorts for medical care or other activities. Support investments to increase routine staffing capacity during the trial period if necessary.

2. Strengthen education about TB and vaccines to facilitate voluntary, informed consent

Develop consent processes tailored to the prison context, ensuring clarity, autonomy, and the absence of coercion. Information must be accessible, culturally appropriate, and delivered in plain language. Provide additional, independent educational resources about TB, vaccines, and clinical trials to all PDL to enable informed, conscious deliberation around the decision to participate. Highlight that participation is voluntary and that one's decision to participate will not affect other rights or privileges in the prison.
Actionable Steps:

- Create a community advisory board with PDL to review and refine consent processes, informational handouts, and educational resources.

- Begin educational activities about TB, vaccines, and clinical trials in months (or years) prior to recruitment, and evaluate improvements in knowledge and understanding prior to initiating trial enrollment.

- Include in the consent form a signed, legally binding assurance from the prison administration that refusal to participate will not affect sentencing, parole, visitation rights, or any other aspect of incarceration.

- Consider incorporating a "teach back" [30] component in the consent process where PDL explains the study purpose and risks to the researcher, to ensure conscious and informed participation.

3. Enact parallel efforts to mitigate TB vulnerability

Evaluate how the heightened vulnerability to TB and fears of infection among PDL may influence their capacity to genuinely weigh risks and benefits when deciding to participate. Incorporate additional safeguards to prevent exploitation of these vulnerabilities, including implementing parallel, prison-wide measures to reduce TB transmission.
Actionable Steps:

• Implement facility-wide active TB screening and treatment for all PDL (regardless of trial participation) and staff periodically (every 4–12 months) before, during, and after the trial.

• Establish a comprehensive, transparent, and visible protocol for TB treatment and follow-up to prevent adverse outcomes and reduce distress and fear around the disease.

4. Actively involve PDL in study design and oversight

Incorporate the voices and lived experiences of PDL in all phases of the study, especially the design and implementation phases. Structured mechanisms such as community advisory boards, as well as more informal processes like open forums or discussion circles, serve to promote transparency, trust, and accountability. During the active phase of the study, these mechanisms also provide an ongoing channel of communication to inform stakeholders about study progress and results, proactively address concerns, and seek feedback and input for protocol modifications if necessary.
Actionable Steps:

• Develop a community advisory board (CAB) with PDL in every study site, with formalized veto power over key operational aspects of the study.

• Hold regular forums with PDL across pavilions/housing units before, during, and after the study.

• Include PDL in dissemination of results from the study and keep them informed about advances as a result of the study

5. Guarantee post-trial access to effective vaccines

If a vaccine candidate is proven safe and effective, all participants must be granted priority access to the new TB vaccine once approved. Detailed vaccine allocation policies and logistical frameworks must be established in advance to ensure timely access for PDL to a successful vaccine following trial completion.
Actionable Steps:

• Develop a formal, legally binding commitment, with parties involved in vaccine allocation (i.e., ministry of health, vaccine manufacturer), to prioritize PDL for access once the vaccine is available. Include prison staff and visitors if possible.

Secure funding for post-trial vaccine distribution in prisons from the study sponsor, including support for logistical and staffing needs and informational resources and campaigns.

6. Recognize the autonomy and right of PDL to participate in science

Respect the right of PDL to make informed decisions about their own health and to access scientific advancements, including through participation in research. While imprisonment may entail the loss of certain rights, it should not automatically preclude PDL from exercising their autonomy in deciding whether to participate in research. Ethical challenges surrounding the inclusion of PDL in clinical trials–rather than serving as blanket justifications for exclusion–should be met with comprehensive measures, strong safeguards, and effective oversight mechanisms developed in partnership with PDL and their advocates.

Actionable Steps:

- Recognizing that priorities, perspectives, and experiences may differ across countries and cultures, center the voices of local PDL in developing ethical frameworks and guidelines that govern research in prisons, balancing the right to participate in and benefit from science with the ethical complexities of carceral settings.

- In partnership with PDL, develop and evaluate "model" strategies for mitigating coercion and enhancing autonomy for research in carceral settings.

## Discussion

In ethical debates around the inclusion of PDL in vaccine trials for TB and other diseases, it is crucial to center the perspectives of those with lived experience of incarceration. However, these perspectives remain poorly understood, particularly in low- and middle-income countries such as Brazil. This qualitative study aimed to understand the perceptions and experiences of PDL across Brazil regarding TB, new TB vaccines, and their willingness to participate in vaccine trials.

Participants expressed a strong desire to protect themselves, their families, and their communities from TB, recognizing both the severity of the disease and the importance of vaccination as a preventive measure. Willingness to participate in TB vaccine trials was conditional on clear and transparent communication and strong safeguards to allow autonomous participation free from coercion. This aligns with the global scoping review on vaccination among People Deprived of Liberty (PDL) [31], which highlights the need for voluntary, trust-based, and contextually adapted immunization strategies within prison settings. The convergence between our findings and these international recommendations reinforces the principle that vaccination initiatives in prisons should prioritize autonomy, informed choice, and continuity of care, while addressing structural barriers that perpetuate health inequities.

Empirical data already demonstrate vaccine acceptance and efficacy in prison settings, as a study examining viral respiratory infections in an Australian maximum-security prison found that SARS-CoV-2 vaccination conferred substantial protection despite the high risk of transmission [32]. Specifically, the incidence of infection was halved (adjusted hazard ratio of 0.46) in fully vaccinated individuals, demonstrating that immunization remains highly effective even under conditions of extreme overcrowding and viral introduction [32]. Positive results were also demonstrated in the study on the efficacy of the mRNA-1273 vaccine during a SARS-CoV-2 Delta outbreak in a US prison [33]. This evidence strongly supports the effectiveness of vaccines in carceral settings and the potential public health benefit of including PDLs in future tuberculosis vaccine trials.

Notably, participants defended their ability to make conscious decisions around participation in research and declared their right to access participation in science. They pointed to their experience with the present study as an example of research that respected their autonomy through transparency, clear communication, and the freedom to decline participation. They also highlighted the importance of active engagement of PDL throughout the research process and the need for clear mechanisms for protection and recourse in case of harm. These findings suggest that, despite the inherent vulnerabilities associated with incarceration, PDL view their inclusion in research as ethically possible and a right to which they are entitled, provided that rigorous safeguards are upheld.

A complicating factor is the current state of prison healthcare in Brazilian prisons. Participants reported difficulties accessing essential care and medicines, indifference and neglect from prison staff, a lack of reliable information, and significant fears of TB. Similar findings of inadequate healthcare services and systemic neglect have been reported in surveys from prisons throughout the region [8,34]. These findings underscore the necessity of improving healthcare access for all PDL to prevent coercive influences on research participation and to ensure quality care before, during, and after a research study.

The coexistence of enthusiasm and skepticism toward research participation underscores the ambivalent nature of trust within carceral settings. While many participants recognized the social and scientific value of clinical research, others

interpreted it through a lens of vulnerability and historical neglect. Distrust was not absolute but conditional—shaped by the degree of transparency, voluntariness, and external oversight. In particular, participants tended to express greater trust in external research teams than in prison health authorities, likely reflecting previous experiences of neglect or coercion in routine health care and vaccination campaigns. This distinction has important ethical implications: ensuring that vaccine trials are clearly led by independent research teams, rather than correctional staff, may enhance their perceived legitimacy and acceptability. Moreover, fears of being used as test subjects reveal legitimate concerns about power imbalances and exploitation. Addressing these perceptions through participatory communication, trustworthy partnerships, and transparent ethical engagement is essential to fostering genuine autonomy and informed consent in future vaccine research involving PDL.

Relatedly, we found pervasive fears of TB and limited access to TB information and services, which may generate pressure to participate in trials due to the absence of alternative preventive measures. To mitigate this potential source of coercion, it is essential to implement parallel efforts to reduce the exorbitant TB risk in prisons. Furthermore, the lack of accurate and accessible information about TB must be addressed to enable PDL to fully and appropriately weigh the risks and benefits of trial participation. Educational initiatives led by trusted health professionals—identified in other prison-based studies as critical for fostering informed decision-making and building trust—are likely to be essential [35–37].

While incarceration entails certain restrictions, the right to health—enshrined in international human rights frameworks—must be fully upheld [19]. This includes not only access to TB care and services but also the right to participate in scientific research and benefit from its advances. The United Nations Mandela Rules affirm that PDL retain all fundamental rights, including access to health and scientific progress, consistent with human dignity [38]. The right to science, as outlined in international covenants such as the ICESCR (Article 15), includes both the right to benefit from scientific advancement and to participate in its development [39].To date, however, participation in research has been severely restricted for PDL due to carceral conditions that impede voluntary, informed, and ethically sound engagement. As discussed in **our recently published perspective in** *The Lancet Infectious Diseases* [40], this exclusion reinforces systemic inequities in who has access to medical advancements and contributes to the invisibility of vulnerable populations in scientific agendas. These barriers, though real, must not be used as justification for the continued systematic exclusion of incarcerated individuals from science. Instead, they underscore the need to build frameworks that enable ethical inclusion with transparency, autonomy, and protection. The recommendations outlined in this study, informed by the perspectives and attitudes of PDL, provide guidance for future strategies to enable the ethical inclusion of PDL in TB vaccine trials, respecting their autonomy and upholding their right to participate in scientific research [39].This study has several limitations. Because participants were recruited through a multi-stage convenience sampling strategy and pre-identified by prison social service teams, selection bias may have influenced who participated and, consequently, the perspectives represented. However, comparing the narratives obtained across all participating prison units, we observed strong thematic convergence across study sites, suggesting that the perspectives captured were not subgroup-specific but rather reflected shared experiences and viewpoints across contexts. Our results may nonetheless have limited generalizability to settings with different TB epidemiology, prison conditions, and cultural contexts. Future studies may explore these topics in surveys with larger, more heterogeneous samples and settings.

We also acknowledge the critical power asymmetry inherent in research involving PDL. The institutional affiliations, social identities, and beliefs of the researchers, especially the facilitators, may have influenced participant responses during focus groups. To mitigate this, facilitators with demonstrated expertise in this methodology posed questions carefully, aiming to avoid guiding questions or value judgments that may have influenced responses. Moreover, facilitators intentionally probed for divergent beliefs and provided strong assurances of confidentiality to encourage sharing of honest, authentic perspectives. Despite these attempts, our findings may have still been subject to social desirability bias, as participants might have felt compelled to offer responses that aligned with perceived expectations of facilitators or peers. The group setting may also have constrained open discussion on sensitive topics, including potential experiences of coercion, distrust, or pressure from prison authorities, gang dynamics, or other PDL. In highly stratified prison environments, group

hierarchies and social tensions may influence the expression of dissent, fear, or more critical perspectives on participation in vaccine trials. Future studies could consider using complementary methodologies—such as anonymous surveys or in-depth individual interviews—to elicit more candid responses and triangulate findings across formats.

## Conclusion

This study investigated the perceptions of PDL in Brazil regarding research on TB and vaccines, aiming to understand their knowledge, experiences, and challenges in this context. Our qualitative analysis revealed several potential sources of structural coercion for vaccine trials in prisons, including unmet health needs, prevalent fears of TB, neglect by prison staff, and limited access to information. Nonetheless, many individuals expressed interest in participating in TB vaccine trials, conditional upon clear, transparent information, engagement of PDL, protection from coercion, and recourse in case of harm. Comprehensive efforts to fulfill these conditions are urgently needed to enable the ethical inclusion of PDL in clinical trials and to promote their equitable access to scientific innovations.

Building on these findings, and to ensure the ethical validity and integrity of future research, we strongly recommend adopting a comprehensive framework of safeguards in prison settings. This framework must prioritize eliminating structural coercion by guaranteeing universal access to quality healthcare for all PDL, independent of trial participation. Crucially, it must mandate robust, multi-format informed consent processes and actively integrate the PDL's voice into study design and oversight. Parallel efforts to mitigate the high vulnerability to TB within the facility are also essential to reduce the coercive influence of disease fear. Ultimately, upholding the autonomy and right of PDL to participate in science, coupled with a legally binding commitment to post-trial access to successful vaccines, is necessary to transform ethical challenges into a model of equitable scientific inclusion.

## Supporting information

**S1 Data. Complete Dataset Supporting the Study Findings.**
(DOCX)

**S1 Table. Guiding Questions for Focus Group Discussions on Health Care, Tuberculosis, and Acceptability of a New TB Vaccine among Incarcerated Individuals.**
(DOCX)

**S2 Table. Distribution of Narratives by Core Theme, Subcategory, State, and Prison Type.**
(DOCX)

**S3 Table. Matrix of Narrative Summary: Convergence and Divergence by Core Theme.**
(DOCX)

## Author contributions

**Conceptualization:** Jason R. Andrews, Crhistinne Cavalheiro Maymone Gonçalves, Julio Croda.

**Data curation:** Mariana Cristina Campos Falleiros Pires, Yiran E. Liu, Everton Ferreira Lemos, Liliane Ferreira da Silva, Julio Croda.

**Formal analysis:** Mariana Cristina Campos Falleiros Pires, Yiran E. Liu, Everton Ferreira Lemos, Mariana Garcia Croda, Julio Croda.

**Funding acquisition:** Jason R. Andrews, Julio Croda.

**Investigation:** Mariana Cristina Campos Falleiros Pires, Yiran E. Liu, Everton Ferreira Lemos, Liliane Ferreira da Silva, Mariana Garcia Croda, Monica Magalhães, Dhélio Batista Pereira, Mariana Pinheiro Alves Vasconcelos, Solana

Monteiro Batista, Giselle Lima de Freitas, Marcelo Cordeiro-Santos, Giane Zupellari dos Santos-Melo, Jair dos Santos Pinheiro, Lara B. O. Assis, Lia Gonçalves Possuelo, Tiago Antonio Heringer, Daiane Kist Back, Pauline Schwarzbold, Julio Croda.

**Methodology:** Everton Ferreira Lemos, Liliane Ferreira da Silva, Jason R. Andrews, Crhistinne Cavalheiro Maymone Gonçalves, Julio Croda.

**Resources:** Everton Ferreira Lemos, Jason R. Andrews, Crhistinne Cavalheiro Maymone Gonçalves, Julio Croda.

**Supervision:** Mariana Cristina Campos Falleiros Pires, Yiran E. Liu, Everton Ferreira Lemos, Liliane Ferreira da Silva, Mariana Garcia Croda, Monica Magalhães, Dhélio Batista Pereira, Mariana Pinheiro Alves Vasconcelos, Solana Monteiro Batista, Giselle Lima de Freitas, Marcelo Cordeiro-Santos, Giane Zupellari dos Santos-Melo, Jair dos Santos Pinheiro, Lara B. O. Assis, Lia Gonçalves Possuelo, Tiago Antonio Heringer, Daiane Kist Back, Pauline Schwarzbold, Jason R. Andrews, Crhistinne Cavalheiro Maymone Gonçalves, Julio Croda.

**Validation:** Yiran E. Liu, Liliane Ferreira da Silva, Monica Magalhães, Rosilene Ruffato, Tiago Antonio Heringer, Jason R. Andrews, Crhistinne Cavalheiro Maymone Gonçalves, Julio Croda.

**Visualization:** Mariana Cristina Campos Falleiros Pires, Yiran E. Liu, Everton Ferreira Lemos, Liliane Ferreira da Silva, Mariana Garcia Croda, Monica Magalhães, Dhélio Batista Pereira, Mariana Pinheiro Alves Vasconcelos, Rosilene Ruffato, Solana Monteiro Batista, Giselle Lima de Freitas, Marcelo Cordeiro-Santos, Giane Zupellari dos Santos-Melo, Jair dos Santos Pinheiro, Lara B. O. Assis, Lia Gonçalves Possuelo, Tiago Antonio Heringer, Daiane Kist Back, Pauline Schwarzbold, Jason R. Andrews, Crhistinne Cavalheiro Maymone Gonçalves, Julio Croda.

**Writing – original draft:** Mariana Cristina Campos Falleiros Pires, Yiran E. Liu, Everton Ferreira Lemos, Liliane Ferreira da Silva, Mariana Garcia Croda, Monica Magalhães, Dhélio Batista Pereira, Mariana Pinheiro Alves Vasconcelos, Rosilene Ruffato, Solana Monteiro Batista, Giselle Lima de Freitas, Marcelo Cordeiro-Santos, Giane Zupellari dos Santos-Melo, Jair dos Santos Pinheiro, Lara B. O. Assis, Lia Gonçalves Possuelo, Tiago Antonio Heringer, Daiane Kist Back, Pauline Schwarzbold, Jason R. Andrews, Crhistinne Cavalheiro Maymone Gonçalves, Julio Croda.

**Writing – review & editing:** Mariana Cristina Campos Falleiros Pires, Yiran E. Liu, Everton Ferreira Lemos, Liliane Ferreira da Silva, Mariana Garcia Croda, Monica Magalhães, Dhélio Batista Pereira, Mariana Pinheiro Alves Vasconcelos, Rosilene Ruffato, Solana Monteiro Batista, Giselle Lima de Freitas, Marcelo Cordeiro-Santos, Giane Zupellari dos Santos-Melo, Jair dos Santos Pinheiro, Lara B. O. Assis, Lia Gonçalves Possuelo, Tiago Antonio Heringer, Daiane Kist Back, Pauline Schwarzbold, Jason R. Andrews, Crhistinne Cavalheiro Maymone Gonçalves, Julio Croda.

## Acknowledgments

The authors thank the people deprived of liberty who agreed to participate in the study, to the state health and public security departments and research centers for their full support during the study period.

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
