## [Decision Letter · Decision Letter 0]

22 Sep 2025

PGPH-D-25-01609

Perceptions of persons deprived of liberty regarding tuberculosis vaccine research

Dear Dr. %Croda%,

Thank you for submitting your manuscript to PLOS Global Public Health. After careful consideration, we feel that it has merit but does not fully meet PLOS Global Public Health’s publication criteria as it currently stands. Therefore, we invite you to submit a revised version of the manuscript that addresses the points raised during the review process.

We look forward to receiving your revised manuscript.

Kind regards,

Zulma Vanessa Rueda, M.D. Ph.D.

Academic Editor

Journal Requirements:

1. Please clarify all sources of funding (financial or material support) for your study. List the grants (with grant number) or organizations (with url) that supported your study, including funding received from your institution. 

2. State the initials, alongside each funding source, of each author to receive each grant.

3. State what role the funders took in the study. If the funders had no role in your study, please state: “The funders had no role in study design, data collection and analysis, decision to publish, or preparation of the manuscript.”

4. If any authors received a salary from any of your funders, please state which authors and which funders.

2. In the online submission form, you indicated that The data that support the findings of this study are available from the corresponding author upon reasonable request, in accordance with ethical guidelines and participant confidentiality. 

3. Uploaded as supplementary information.

3. Please provide separate figure files in .tif or .eps format.

4. Please insert an Ethics Statement at the beginning of your Methods section, under a subheading 'Ethics Statement'. It must include:

1) The name(s) of the Institutional Review Board(s) or Ethics Committee(s)

2) The approval number(s), or a statement that approval was granted by the named board(s) 

3) (for human participants/donors) - A statement that formal consent was obtained (must state whether verbal/written) OR the reason consent was not obtained (e.g. anonymity). NOTE: If child participants, the statement must declare that formal consent was obtained from the parent/guardian.

5. We do not publish any copyright or trademark symbols that usually accompany proprietary names, eg (R), (C), or TM  (e.g. next to drug or reagent names). Please remove all instances of trademark/copyright symbols throughout the text, including ® on page 7, 8.

6. Some material included in your submission may be copyrighted. According to PLOS’s copyright policy, authors who use figures or other material (e.g., graphics, clipart, maps) from another author or copyright holder must demonstrate or obtain permission to publish this material under the Creative Commons Attribution 4.0 International (CC BY 4.0) License used by PLOS journals. Please closely review the details of PLOS’s copyright requirements here: PLOS Licenses and Copyright. If you need to request permissions from a copyright holder, you may use PLOS's Copyright Content Permission form.

Potential Copyright Issues:

Figure 1: please (a) provide a direct link to the base layer of the map (i.e., the country or region border shape) and ensure this is also included in the figure legend; and (b) provide a link to the terms of use / license information for the base layer image or shapefile. We cannot publish proprietary or copyrighted maps (e.g. Google Maps, Mapquest) and the terms of use for your map base layer must be compatible with our CC-BY 4.0 license. 

7. We notice that your supplementary tables are included in the manuscript file. Please remove them and upload them with the file type 'Supporting Information'. Please ensure that each Supporting Information file has a legend listed in the manuscript after the references list.

Additional Editor Comments:

Reviewer #1:

This is a relevant and timely study that makes a valuable contribution to the literature on research ethics and tuberculosis control among people in prison.

Minor comments:

Under "Respect for autonomy," the bracketed note "[what is 'it' referring to? ...]" should be removed from the final publication version as it appears to be an editorial artifact.

The discussion could be strengthened by linking participants’ expressed willingness to vaccination with empirical evidence that incarcerated populations both accept and benefit from vaccination. For example, this recent paper in a maximum-security prison found that SARS-CoV-2 vaccination conferred substantial protection mid-outbreak, despite overcrowding and multiple viral introductions (DOI: 10.3201/eid3108.240571). Citing evidence from other real-world scenarios and infections would reinforce the finding that PDL are not inherently resistant to vaccination and that vaccination remains effective, even under extreme transmission conditions. This would underscore the relevance and public health good of including PDL in TB (and other) vaccine trials.

The recommendations, in their current form are a bit too abstract to serve as a practical guide, and read as a conclusion rather than a tool for action. The authors are uniquely positioned to leverage their empirical findings to propose concrete and actionable operational frameworks. I urge them to revise this section to provide the specific, nitty-gritty details that would transform these principles from admirable goals into a usable blueprint for ethical practice.

e.g. The advice to "tailor" consent is generic. What does a prison-specific consent process concretely look like in terms of literacy level, cultural format, and verification checkpoints to ensure it is a truly ongoing process?

The manuscript notes that some individuals refused participation but does not explore or report their reasons. In a study fundamentally concerned with autonomy and coercion in a carceral setting, understanding the perspectives of those who declined is critical. Being transparent about their reasons would provide a more complete understanding of the landscape of opinion and the potential barriers to ethical recruitment.

Similarly, a more explicit and reflexive discussion of researcher positionality is required. The power differential between academic researchers and incarcerated participants is the central context of this work. The manuscript would be strengthened by a critical examination of how the researchers' social identities, institutional affiliations, and preconceptions may have influenced focus group facilitation, participant responses, and data interpretation.

Major comments:

One thing that appears conspicuous is the absence of discordant perspectives, or any discussion of dissent, disagreement, or negative cases. Were there really no participants who expressed cynicism or distrust? The results present a consensus, supported by illustrative quotes, with no outliers that challenge the manuscripts narrative. If there were participants who expressed strong opposition, or distrust regarding research participation, these perspectives should be integrated into the analysis. Their absence or omission risks presenting an overly consensual and potentially incomplete picture.

On a related note, the use of convenience sampling is mentioned almost in passing, and requires elaboration. The manuscript does not state how many individuals initially volunteered, nor whether any were excluded after volunteering. How were participants actually selected? Did everyone have the opportunity to volunteer? Were they pre-vetted? Were they considered the more compliant, less fearful, or more privileged individuals within the prison hierarchy? The potential for selection bias here is substantial and fundamentally shapes who gets to have a "perspective" in this study. Those who volunteered may represent a specific subset of the population with shared traits or living conditions not representative of the general population. In a multi-site study across diverse prisons, not providing more detail here is a unignorable weakness.

Reviewer #2:

This is a timely and important manuscript addressing the ethical and practical considerations of including persons deprived of liberty in tuberculosis vaccine trials. The study is methodologically sound, and the findings provide valuable insights into participants’ perceptions and concerns, with clear implications for public health policy and research ethics.

Strengths:

• Novel and relevant topic, addressing a major gap in TB research.

• Strong qualitative methodology with triangulation and multi-site sampling.

• Ethical approvals clearly described.

• Well-supported conclusions and practical recommendations grounded in participant voices.

Points for improvement:

1. Data availability: While confidentiality concerns are valid, consider providing an anonymized coded dataset or expanded supplementary material (e.g., summary matrices) to strengthen compliance with PLOS’s data policy.

2. Quotations: Some direct quotations are lengthy or repetitive; consider condensing for readability without losing participants’ perspectives.

3. References: A few references are incomplete (e.g., Ako 2020; Furin, Cox, Pai 2019) and should be corrected/standardized.

4. Clarity in Results: In one passage, the phrase “It’s mandatory to take it” is ambiguous (COVID-19 vaccine vs. TB trial participation). Clarify context.

Overall, this is a strong and ethically rigorous manuscript that will make an important contribution to the field. With minor revisions, it is suitable for publication.

Critical Assessment of the Article:

Perceptions of persons deprived of liberty regarding tuberculosis vaccine research

Strengths

• Novel and highly relevant topic (perceptions of persons deprived of liberty—PDL— regarding TB vaccine trials).

• Well-described qualitative methodology (focus groups, inclusion criteria, thematic analysis with triangulation).

• Inclusion of 91 participants across 7 prisons in 5 Brazilian states, providing diversity.

• Strong discussion, linking findings to international ethical frameworks and recent literature on vulnerability and the right to science.

• Practical recommendations (6 points) are clear and directly derived from participants’ testimonies.

Weaknesses / Areas for Improvement

• Sampling limitations: Convenience sampling was used. The discussion acknowledges this, but could expand on how it affects transferability of results.

• Representation of female voices: Women were included (30%), but most quotations highlighted are from men. Balancing quotes would strengthen the narrative.

• Confusion in one passage: (e.g., “It’s mandatory to take it…”—unclear whether referring to COVID-19 vaccination or TB trial participation). This needs clarification.

• Quotations: Some participant quotes are long and repetitive; condensing would improve readability.

• References: Correct incomplete entries (Ako 2020, Furin 2019, etc.) and standardize formatting.

1. Does this manuscript meet PLOS Global Public Health’s publication criteria? Is the manuscript technically sound, and do the data support the

conclusions?

Yes. The manuscript is technically sound and methodologically rigorous within the qualitative research paradigm. The use of focus groups across multiple prisons, with

systematic thematic content analysis, is appropriate for the research question. Ethical approvals are clearly documented. The conclusions are well aligned with and supported by

the data presented.

2. Has the statistical analysis been performed appropriately and rigorously?

Yes. This is a qualitative study. Quantitative analysis is limited to descriptive statistics (counts, percentages, mean age) to contextualize the sample, which is appropriate. The

qualitative analysis was conducted rigorously, including verbatim transcription, double coding, thematic categorization, and triangulation. No advanced inferential statistics were

required or expected for the study design.

3. Have the authors made all data underlying the findings in their manuscript fully available?

Partially. The Data Availability Statement indicates that data are available upon reasonable request from the corresponding author, due to ethical and confidentiality concerns involving

incarcerated participants. While this justification is valid, it does not fully comply with PLOS’s policy requiring unrestricted availability. The authors may wish to strengthen

compliance by providing anonymized coded data, excerpts, or aggregated datasets that do not compromise participant confidentiality.

4. Is the manuscript presented in an intelligible fashion and written in standard English?

Yes. The manuscript is clearly written, logically structured, and presented in professional English. Participant quotations are used effectively, though some could be shortened for

readability. Minor stylistic editing (condensing lengthy sentences, polishing phrasing in the Discussion) would improve clarity, but overall the manuscript is well-written and suitable for

publication.

5. Review Comments to the Author

This is a timely and important manuscript addressing the ethical and practical considerations of including persons deprived of liberty in tuberculosis vaccine trials. The study is

methodologically sound, and the findings provide valuable insights into participants’ perceptions and concerns, with clear implications for public health policy and research ethics.

Strengths:

• Novel and relevant topic, addressing a major gap in TB research.

• Strong qualitative methodology with triangulation and multi-site sampling.

• Ethical approvals clearly described.

• Well-supported conclusions and practical recommendations grounded in participant voices.

Points for improvement:

1. Data availability: While confidentiality concerns are valid, consider providing an anonymized coded dataset or expanded supplementary material (e.g., summary

matrices) to strengthen compliance with PLOS’s data policy.

2. Quotations: Some direct quotations are lengthy or repetitive; consider condensing for readability without losing participants’ perspectives.

3. References: A few references are incomplete (e.g., Ako 2020; Furin, Cox, Pai 2019) and should be corrected/standardized.

4. Clarity in Results: In one passage, the phrase “It’s mandatory to take it” is ambiguous (COVID-19 vaccine vs. TB trial participation). Clarify context.

Overall, this is a strong and ethically rigorous manuscript that will make an important contribution to the field. With minor revisions, it is suitable for publication

Reviewers' comments:

Reviewer's Responses to Questions

**Comments to the Author**

1. Does this manuscript meet PLOS Global Public Health’s publication criteria?

Reviewer #1: Partly

Reviewer #2: Yes

2. Has the statistical analysis been performed appropriately and rigorously?

Reviewer #1: N/A

Reviewer #2: N/A

3. Have the authors made all data underlying the findings in their manuscript fully available (please refer to the Data Availability Statement at the start of the manuscript PDF file)?

Reviewer #1: No

Reviewer #2: No

4. Is the manuscript presented in an intelligible fashion and written in standard English?

Reviewer #1: Yes

Reviewer #2: Yes

Reviewer #1: This is a relevant and timely study that makes a valuable contribution to the literature on research ethics and tuberculosis control among people in prison.

Minor comments:

Under "Respect for autonomy," the bracketed note "[what is 'it' referring to? ...]" should be removed from the final publication version as it appears to be an editorial artifact.

The discussion could be strengthened by linking participants’ expressed willingness to vaccination with empirical evidence that incarcerated populations both accept and benefit from vaccination. For example, this recent paper in a maximum-security prison found that SARS-CoV-2 vaccination conferred substantial protection mid-outbreak, despite overcrowding and multiple viral introductions (DOI: 10.3201/eid3108.240571). Citing evidence from other real-world scenarios and infections would reinforce the finding that PDL are not inherently resistant to vaccination and that vaccination remains effective, even under extreme transmission conditions. This would underscore the relevance and public health good of including PDL in TB (and other) vaccine trials.

The recommendations, in their current form are a bit too abstract to serve as a practical guide, and read as a conclusion rather than a tool for action. The authors are uniquely positioned to leverage their empirical findings to propose concrete and actionable operational frameworks. I urge them to revise this section to provide the specific, nitty-gritty details that would transform these principles from admirable goals into a usable blueprint for ethical practice.

e.g. The advice to "tailor" consent is generic. What does a prison-specific consent process concretely look like in terms of literacy level, cultural format, and verification checkpoints to ensure it is a truly ongoing process?

The manuscript notes that some individuals refused participation but does not explore or report their reasons. In a study fundamentally concerned with autonomy and coercion in a carceral setting, understanding the perspectives of those who declined is critical. Being transparent about their reasons would provide a more complete understanding of the landscape of opinion and the potential barriers to ethical recruitment.

Similarly, a more explicit and reflexive discussion of researcher positionality is required. The power differential between academic researchers and incarcerated participants is the central context of this work. The manuscript would be strengthened by a critical examination of how the researchers' social identities, institutional affiliations, and preconceptions may have influenced focus group facilitation, participant responses, and data interpretation.

Major comments:

One thing that appears conspicuous is the absence of discordant perspectives, or any discussion of dissent, disagreement, or negative cases. Were there really no participants who expressed cynicism or distrust? The results present a consensus, supported by illustrative quotes, with no outliers that challenge the manuscripts narrative. If there were participants who expressed strong opposition, or distrust regarding research participation, these perspectives should be integrated into the analysis. Their absence or omission risks presenting an overly consensual and potentially incomplete picture.

On a related note, the use of convenience sampling is mentioned almost in passing, and requires elaboration. The manuscript does not state how many individuals initially volunteered, nor whether any were excluded after volunteering. How were participants actually selected? Did everyone have the opportunity to volunteer? Were they pre-vetted? Were they considered the more compliant, less fearful, or more privileged individuals within the prison hierarchy? The potential for selection bias here is substantial and fundamentally shapes who gets to have a "perspective" in this study. Those who volunteered may represent a specific subset of the population with shared traits or living conditions not representative of the general population. In a multi-site study across diverse prisons, not providing more detail here is a unignorable weakness.

Reviewer #2: This is a timely and important manuscript addressing the ethical and practical considerations of including persons deprived of liberty in tuberculosis vaccine trials. The study is methodologically sound, and the findings provide valuable insights into participants’ perceptions and concerns, with clear implications for public health policy and research ethics.

Strengths:

• Novel and relevant topic, addressing a major gap in TB research.

• Strong qualitative methodology with triangulation and multi-site sampling.

• Ethical approvals clearly described.

• Well-supported conclusions and practical recommendations grounded in participant voices.

Points for improvement:

1. Data availability: While confidentiality concerns are valid, consider providing an anonymized coded dataset or expanded supplementary material (e.g., summary matrices) to strengthen compliance with PLOS’s data policy.

2. Quotations: Some direct quotations are lengthy or repetitive; consider condensing for readability without losing participants’ perspectives.

3. References: A few references are incomplete (e.g., Ako 2020; Furin, Cox, Pai 2019) and should be corrected/standardized.

4. Clarity in Results: In one passage, the phrase “It’s mandatory to take it” is ambiguous (COVID-19 vaccine vs. TB trial participation). Clarify context.

Overall, this is a strong and ethically rigorous manuscript that will make an important contribution to the field. With minor revisions, it is suitable for publication.

**Do you want your identity to be public for this peer review?** For information about this choice, including consent withdrawal, please see our Privacy Policy

Reviewer #1: No

Reviewer #2: No

---

## [Editor Report · Decision Letter 1]

18 Nov 2025

Perceptions of persons deprived of liberty regarding tuberculosis vaccine research

PGPH-D-25-01609R1

Dear %Dr.% %Croda%,

We are pleased to inform you that your manuscript 'Perceptions of persons deprived of liberty regarding tuberculosis vaccine research' has been provisionally accepted for publication in PLOS Global Public Health.

Best regards,

Zulma Vanessa Rueda, M.D. Ph.D.

Academic Editor

Thanks for your work and leadership in this field. The changes suggested by reviewers were incorporated in the resubmitted manuscript and I appreciate the actionable steps that you incorporated as it provides clear suggestions and next steps for researchers working in this field.

This manuscript provides novel and important findings to advance the field of clinical trials among people deprived of liberty with a meaningful participation and considerations before, during and after trials are conducted.

Congratulations to the authors.